# Comparative Efficacy and Safety of P2Y12 Inhibitor Monotherapy and Dual Antiplatelet Therapy in Patients with and without Diabetes Mellitus Undergoing Percutaneous Coronary Intervention

**DOI:** 10.3390/ijms23094549

**Published:** 2022-04-20

**Authors:** Wen-Han Feng, Yong-Chieh Chang, Yi-Hsiung Lin, Hsiao-Ling Chen, Hsiu-Mei Chang, Chih-Sheng Chu

**Affiliations:** 1Department of Internal Medicine, Kaohsiung Municipal Ta-Tung Hospital, Kaohsiung Medical University Hospital, Kaohsiung Medical University, Kaohsiung 80145, Taiwan; hans0426@gmail.com; 2Department of Pharmacy, Kaohsiung Municipal Ta-Tung Hospital, Kaohsiung 80145, Taiwan; lovemjayo@gmail.com (Y.-C.C.); hlchen369@gmail.com (H.-L.C.); 880504@kmhk.org.tw (H.-M.C.); 3Department of Internal Medicine, Division of Cardiology, Kaohsiung Medical University Hospital, Kaohsiung 80756, Taiwan; caminolin@gmail.com; 4Center for Lipid Biosciences, Kaohsiung Medical University Hospital, Kaohsiung 80756, Taiwan

**Keywords:** P2Y12 inhibitor monotherapy, percutaneous coronary intervention (PCI), diabetes mellitus (DM)

## Abstract

Increasing evidence has shown P2Y12 inhibitor monotherapy is a feasible alternative treatment for patients after percutaneous coronary intervention (PCI) with stent implantation in the modern era. However, patients with diabetes mellitus (DM) have a higher risk of ischemic events and more complex coronary artery disease. The purpose of this study is to evaluate the efficacy and safety of this novel approach among patients with DM and those without DM. We conducted a systematic review and meta-analysis of randomized controlled trials that compared P2Y12 inhibitor monotherapy with 12 months of dual antiplatelet therapy (DAPT) in patients who underwent PCI with stent implantation. PubMed, Embase, Cochrane library database, ClinicalTrials.gov, and three other websites were searched for our data from the earliest report to January 2022. The primary efficacy outcome was major adverse cardiovascular and cerebrovascular events (MACCE): a composite of all-cause mortality, myocardial infarction, stent thrombosis, and stroke. The primary safety outcome was major or minor bleeding events. The secondary endpoint was net adverse clinical events (NACE) which are defined as a composite of major bleeding and adverse cardiac and cerebrovascular events. A total of four randomized controlled trials with 29,136 patients were included in our meta-analysis. The quantitative analysis showed a significant reduction in major or minor bleeding events in patients treated with P2Y12 inhibitor monotherapy compared to standard DAPT (OR: 0.68, 95% CI: 0.46–0.99, *p* = 0.04) without increasing the risk of MACCE (OR: 0.96, 95% CI: 0.85–1.09, *p* = 0.50). The number of NACE was significantly lower in the patients treated with P2Y12 inhibitor monotherapy (OR: 0.84, 95% CI: 0.72–0.97, *p* = 0.019). In DM patients, P2Y12 inhibitor monotherapy was associated with a lower risk of MACCE compared to standard DAPT (OR: 0.85, 95% CI: 0.74–0.98, *p* = 0.02). Furthermore, P2Y12 inhibitor monotherapy was accompanied by a favorable reduction in major or minor bleeding events (OR: 0.80, 95% CI: 0.64–1.05, *p* = 0.107). In non-DM patients, P2Y12 inhibitor monotherapy showed a significant reduction in major or minor bleeding events (OR: 0.58, 95% CI: 0.38–0.88, *p* = 0.01), but without increasing the risk of MACCE (OR: 0.99, 95% CI: 0.82–1.19, *p* = 0.89). Based on these findings, P2Y12 inhibitor monotherapy could significantly decrease bleeding events without increasing the risk of stent thrombosis or myocardial infarction in the general population. The benefit of reducing bleeding events was much more significant in non-DM patients than in DM patients. Surprisingly, P2Y12 inhibitor monotherapy could lower the risk of MACCE in DM patients. Our study supports that P2Y12 inhibitor monotherapy is a promising alternative choice of medical treatment for patients with DM undergoing PCI with stent implantation in the modern era.

## 1. Introduction

Dual antiplatelet therapy (DAPT) with aspirin plus a P2Y12 inhibitor is the standard treatment for patients undergoing percutaneous coronary intervention (PCI) with stent implantation [1]. Although it is an effective treatment to reduce the risk of ischemic events and stent thrombosis, it increases the risk of bleeding. Newer-generations of drug-eluting stents (DES) have thinner stent struts and better design to lower the risk of stent thrombosis and have more rapid endothelialization. The role of DAPT was challenged by many clinical trials in recent years [2,3,4,5]. Increasing evidence is showing P2Y12 inhibitor monotherapy is a feasible alternative treatment for patients after PCI with stent implantation in the modern era, as it could lower the risk of bleeding complications and still has enough antiplatelet effect to avoid recurrent ischemic events [6].

Diabetes mellitus (DM) and non-DM patients have very much different clinical characteristics. DM patients are associated with a higher risk of ischemic events and usually have more co-morbidities than non-DM patients [5,7,8]. In addition, DM patients usually have more complex coronary artery disease and more stents implanted during PCI than non-DM patients. Globally, the prevalence of DM has increased significantly in the past decade [9]. Therefore, it is important to find the optimal post-PCI therapy for patients with DM. The efficacy and safety of this novel approach among patients with or without diabetes mellitus is uncertain. Although some clinical trials have shown P2Y12 inhibitor monotherapy had favorable outcomes for DM patients [10,11], they were individually underpowered. Therefore, we perform a systematic review and meta-analysis to assess the efficacy and safety of P2Y12 inhibitor monotherapy compared to DAPT in DM and non-DM patients who underwent PCI and stent implantation.

## 2. Methods

### 2.1. Data Sources and Study Selection

This meta-analysis was conducted following the recommendations of the Preferred Reporting Items for a Systematic review and Meta-analysis of Individual Participant Data (PRISMA-IPD) and the Cochrane Collaboration method. The protocol was registered on PROSPERO (international prospective register of systematic reviews) on 23 February 2022, and is available online (www.crd.york.ac.uk/prospero (accessed on 23 February 2022), CRD42022312669). Starting on the same date, we searched PubMed, Embase, Cochrane library database, ClinicalTrials.gov, and three other websites (www.escardio.org, www.acc.org/cardiosourceplus, www.tctmd.com (accessed on 23 February 2022)) from the earliest record to January 2022. The inclusion criteria of the study were as follows: (1) the study included patients who underwent PCI with stent implantation, (2) the study was a randomized controlled trial comparing P2Y12 inhibitor monotherapy to standard 12-month dual antiplatelet therapy, (3) the study had followed up on patients’ clinical outcomes for at least 12 months after PCI, and (4) the study had reported the primary efficacy and safety outcomes of interest. The search terms used included: “P2Y12 inhibitor monotherapy”, “dual antiplatelet therapy”, “randomized trial”, “percutaneous coronary intervention”, “Outcome”, and “diabetes mellitus”. The exclusion criteria included: (1) non-randomized controlled trial and (2) studies that had not reported the data of patients with DM and non-DM. No language restriction was enforced, and studies not available in full-text were excluded. The detailed search strategies are shown in Appendix A. 

Multiple reviewers examined all the retrieved articles and data using a predetermined form. The quality of each study was evaluated by the first and second authors (Wen-Han Feng and Yong-Chieh Chang) by using the Cochrane Collaboration tool. Discrepancies between the reviewers were solved by discussions with the corresponding author.

### 2.2. Data Extraction and Main Outcomes

The baseline characteristics of included studies were extracted by the first two authors, and the discrepancy was resolved through negotiation. The primary efficacy outcome was major adverse cardiovascular and cerebrovascular events (MACCE); a composite of all-cause mortality, myocardial infarction, stent thrombosis, and stroke. The primary safety outcome was major or minor bleeding events. The secondary endpoint was net adverse clinical events (NACE); defined as a composite of major bleeding and adverse cardiac and cerebrovascular events.

### 2.3. Statistical Analysis

All data were pooled to calculate the hazard ratios and 95% confidence intervals by using a random-effects model. Between-trial heterogeneity was assessed by using an I^2^ test, and if the value was >50% it was regarded as having considerable heterogeneity. Potential publication bias was examined via the visual inspection of funnel plots, Egger’s test, and Begg’s test. Statistical significance is defined as a *p*-value < 0.05. All analyses were performed using Comprehensive Meta-Analysis (CMA) software, version 3 (Biostat, Englewood, NJ, USA).

## 3. Results

### 3.1. Search Results and Study Characteristics

The results of the literature searches and study selections are shown in Figure 1. A total of 2180 records were identified from PubMed, Embase, Cochrane library database, and three other websites (www.escardio.org, www.acc.org/cardiosourceplus, www.tctmd.com, accessed on 23 February 2022). Of these, 28 full-text articles were reviewed, and 24 of them were excluded due to failure to meet the pre-specified inclusion criteria. The STOP-DAPT 2 trial was excluded because there was no available reported data on patients with or without DM to perform the analysis. Finally, four randomized controlled trials were included in this systematic review and meta-analysis.

The main characteristics of included trials are summarized in Table 1. A total of 29,136 patients were available for the primary analysis. There were 8615 patients with diabetes mellitus and 20,507 patients without DM. The ischemic and bleeding events of DM patients and non-DM patients in each trial are summarized in Table 2.

### 3.2. The Primary Efficacy and Safety Outcomes

In overall enrolled patients, the quantitative analysis is demonstrated in Figure 2. There was no increased risk of MACCE in patients treated with P2Y12 inhibitor monotherapy compared to standard 12-month DAPT (OR: 0.96, 95% CI: 0.85–1.09, *p* = 0.50, *I*^2^ = 7%, P_Heterogeneity_ = 0.36), but bleeding events were significantly reduced (OR: 0.68, 95% CI: 0.46–0.99, *p* = 0.04, *I*^2^ = 89%, P_Heterogeneity_ < 0.001). One study that produced heterogeneity was identified via sensitivity analysis, and the heterogeneity was reduced after excluding the results of this trial (*I*^2^ = 0%, P_Heterogeneity_ = 0.81). The net adverse clinical events (a composite endpoint of bleeding and ischemic events) were significantly lower in the patients treated with P2Y12 inhibitor monotherapy (OR: 0.84, 95% CI: 0.72–0.97, *p* = 0.019, *I*^2^ = 40%, P_Heterogeneity_ = 0.17).

The primary efficacy outcomes (a composite of MACCE) of patients with DM and without DM are shown in Figure 3. In DM patients, P2Y12 inhibitor monotherapy significantly lowered the risk of MACCE compared to standard 12-month DAPT (OR: 0.85, 95% CI: 0.74–0.98, *p* = 0.02, *I*^2^ = 0%, P_Heterogeneity_ = 0.62). In non-DM patients, P2Y12 inhibitor monotherapy had a similar risk of MACCE compared to DAPT (OR: 0.99, 95% CI: 0.82–1.19, *p* = 0.89, *I*^2^ = 27%, P_Heterogeneity_ = 0.25).

The primary safety outcomes of patients with DM and without DM are shown in Figure 4. In DM patients, P2Y12 inhibitor monotherapy was associated with a favorable reduction in bleeding events (OR: 0.80, 95% CI: 0.64–1.05, *p* = 0.107, *I*^2^ = 22%, P_Heterogeneity_ = 0.28). In non-DM patients, P2Y12 inhibitor monotherapy showed a great reduction in bleeding events (OR: 0.58, 95% CI: 0.38–0.88, *p* = 0.01, *I*^2^ = 78%, P_Heterogeneity_ = 0.004).

### 3.3. Quality Assessment and Publication Bias

The detailed quality assessment and risk of bias assessment for each study can be found in Appendix A. The overall risk of bias in selection, detection, and reporting bias was low. All studies in this meta-analysis were randomized controlled trials, but only TWILIGHT was double-blinded. There was no publication bias in all outcomes. The outcomes of included trials are distributed symmetrically in the funnel plot (Appendix A), and the *p*-value of the Begg’s and Egger’s tests were more than 0.05 in all outcomes (Appendix A). Heterogeneity was low in all outcomes.

## 4. Discussion

Conventionally, it is recommended that DAPT should be continued for at least 6 months (in stable coronary artery disease) or 12 months (in acute coronary syndrome) unless contraindications occur [12]. Early suspension of antiplatelet therapy would increase the risk of recurrent ischemic events and stent thrombosis. This concept was changed because of the advent of safer, newer-generation DES and the awareness of increased bleeding risk caused by prolonged DAPT. Based on these reasons, a new treatment strategy of using a very short period of DAPT followed by a potent P2Y12 inhibitor monotherapy was proposed. The results from this systematic review and meta-analysis of 29136 patients from four randomized controlled trials indicate that the P2Y12 inhibitor monotherapy could significantly lower the risk of bleeding complications without increasing the risk of ischemic events compared with standard DAPT in patients without DM. Surprisingly, P2Y12 inhibitor monotherapy significantly reduced the risk of ischemic events in patients with DM, but not the risk of bleeding complications. Currently, there is no clear biological rationale to explain this clinical finding. However, ticagrelor seemed to have better clinical effects when combined with a lower dose of aspirin in a PLATO study [13]. One possible hypothesis is that aspirin reduces not only the release of thromboxane A_2_, but also the release of prostacyclin [14]. The therapeutic effect of ticagrelor may be attenuated when endogenous prostacyclin production is inhibited [15,16]. It is possible that ticagrelor works better in monotherapy than in combination therapy with aspirin. Further investigations are needed to elucidate the complex interactions between these two drugs. These findings challenge contemporary practice guideline recommendations for DAPT as the standard treatment for post-PCI care. Other meta-analyses have been published on P2Y12 inhibitor monotherapy after PCI [17,18,19]. However, our meta-analysis is unique in focusing on DM patients. This distinction is important given the growing prevalence of diabetic patients and the different prognostic nature of these patients. The TWILIGHT DM substudy was the very first randomized study to show that P2Y12 inhibitor monotherapy could have better outcomes compared to standard DAPT [10]. However, the case number was relatively small. Our study is the first meta-analysis to show a decrease in the risk of ischemic events with P2Y12 inhibitor monotherapy in patients with DM compared to standard DAPT.

One of the salient findings in our study is that patients with DM indeed had a significantly higher risk of ischemic events after PCI in the modern era. The rate of ischemic events in DM and non-DM patients treated with standard DAPT was 18.7% vs. 12.8% in the GLOBAL LEADERS trial, 3.8% vs. 1.7% in the SMART-CHOICE trial, 5.9% vs. 2.8% in the TWILIGHT trial, and 5.1% vs. 2.7% in the TICO trial. All four clinical trials had the same findings. However, the risk of bleeding complications was only slightly higher in DM patients than in non-DM patients. There are several possible explanations for these findings. First, platelet reactivity was higher in DM patients than in non-DM patients [20]. Second, the turnover rate and the number of reticulated platelets were both higher in DM patients, resulting in more endothelial cell adhesion [21]. Third, DM patients tend to be more resistant to antiplatelet agents [22]. Based on these factors, we could assume that the bleeding risk of DM patients treated with antiplatelet agents would be similar or even lower than non-DM patients. Therefore, DM patients should be treated with a different antiplatelet regimen than non-DM patients [23].

DM also leads to endothelial dysfunction (one of the main pathophysiologic mechanisms associated with cardiovascular disease) and is described as an independent determinant of ischemic heart disease and acute coronary syndrome [24]. Several biochemical pathways have been described to demonstrate the association between endothelial dysfunction and platelet activation, such as nitric oxide (NO) and prostacyclin (PGI_2_) [25]. NO, a well-known atheroprotective and vasodilating substance, may also attenuate platelet activation. Blood vessels of patients with DM have diminished NO production, enhanced NO degradation, and decreased sensitivity to NO. PGI_2_ is another important regulator produced by endothelial cells that inhibits platelet activation through binding to the prostacyclin receptor on platelets. DM is associated with lower levels of prostacyclin synthase in subcutaneous arteries and possibly leads to impaired formation of PGI_2_ [26]. Together, endothelial dysfunction and platelet hyperactivity make DM patients much more susceptible to cardiovascular disease than non-DM patients [27].

The success of P2Y12 inhibitor monotherapy was not a coincidence. Many in vitro and ex vivo investigations have shown that aspirin provided very limited additional platelet inhibition and anti-thrombotic effect to a potent P2Y12 inhibitor [28,29,30]. Of note, most of the patients treated with P2Y12 inhibitor monotherapy were using ticagrelor. Ticagrelor is a more potent P2Y12 inhibitor than clopidogrel and may improve endothelial function and blood viscosity [31,32]. In addition, clopidogrel is a prodrug that requires metabolism to transform into an active form. DM patients have a greater prevalence of being unresponsive to clopidogrel than non-DM patients [33]. Impaired drug metabolism, metabolic disorders, and competition for CYP3A4 with other drugs (e.g., statins) are possible mechanisms leading to a lower concentration of clopidogrel’s active metabolite and insufficient antiplatelet effects [34,35]. Therefore, it is possible the benefit of P2Y12 inhibitor monotherapy in patients who underwent PCI belongs to ticagrelor alone. In our previous real-world observational study, ticagrelor monotherapy resulted in substantially lower cardiovascular risk compared to clopidogrel monotherapy in patients with acute coronary syndrome (ACS) undergoing PCI [36].

## 5. Limitations

There are several limitations in our study. First, most patients enrolled in these included trials were implanted with newer-generation DESs. Although it is widely used in our daily practice, our findings may not apply to first-generation DESs or bare-metal stents. Second, there were some differences in baseline characteristics and the indications for PCI in included trials. Moreover, only one trial was double-blinded, and others were open-label. Third, there are heterogeneities in the definition of bleeding complications. Although BARC (Bleeding Academic Research Consortium) 3 or 5 is similar to TIMI (Thrombolysis in Myocardial Infarction) minor or major bleeding, they are not identical. Fourth, most of the patients treated with P2Y12 inhibitor monotherapy were using ticagrelor. The outcomes may not apply to other P2Y12 inhibitors.

## 6. Conclusions

Based on this systematic review and meta-analysis, P2Y12 inhibitor monotherapy followed by a short duration of dual antiplatelet therapy could significantly decrease the risk of bleeding events without increasing the risk of stent thrombosis or myocardial infarction in the general population. The benefit of reducing bleeding events was much more significant in non-DM patients than in DM patients. Surprisingly, P2Y12 inhibitor monotherapy could lower the risk of MACCE in DM patients but not in non-DM patients. These findings support that P2Y12 inhibitor monotherapy is a feasible, alternative choice of medical treatment for patients with or without diabetes mellitus undergoing percutaneous intervention with stent implantation in the modern era.

## Figures and Tables

**Figure 1 ijms-23-04549-f001:**
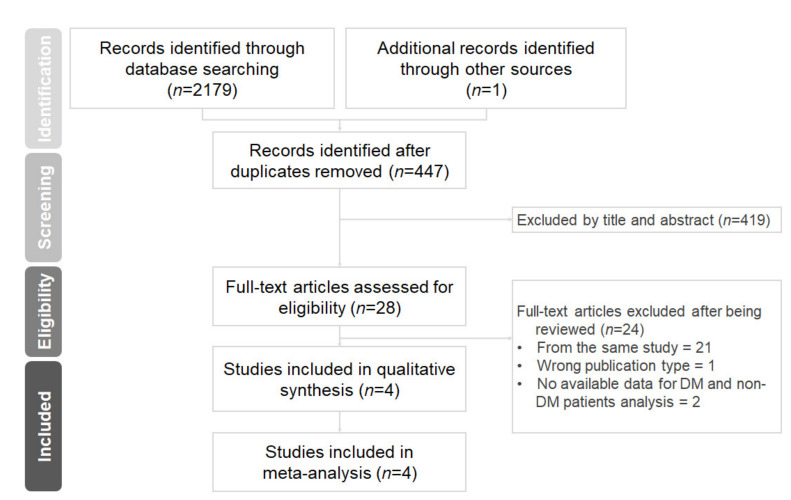
Preferred Reporting Items for Systematic Reviews and Meta-Analyses (PRISMA) diagram for the searching and identification of included studies.

**Figure 2 ijms-23-04549-f002:**
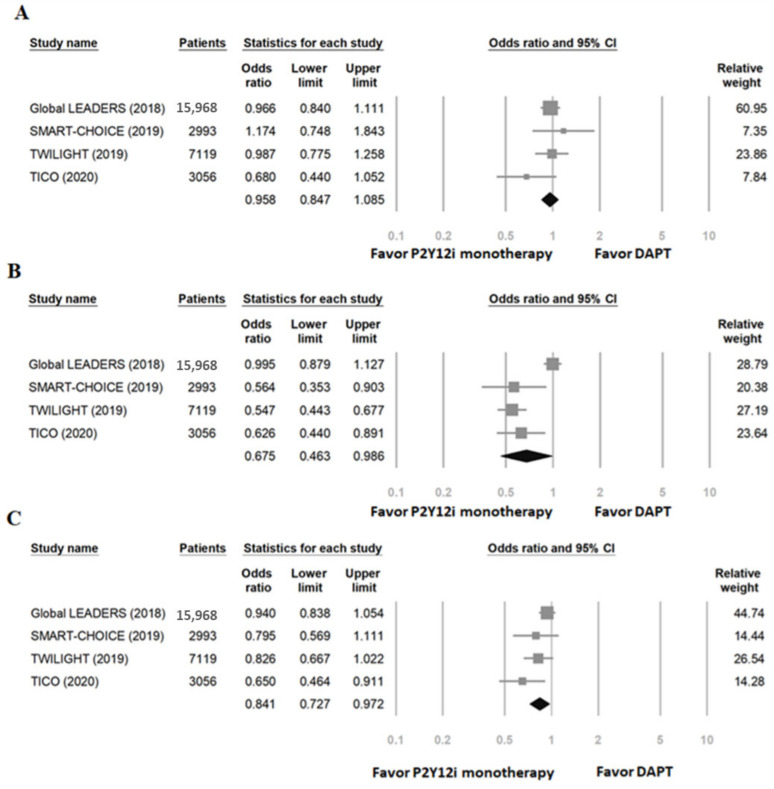
The primary efficacy and safety of P2Y12 inhibitor monotherapy in patients undergoing PCI compared to 12-month DAPT. (**A**) MACCE; (**B**) bleeding events; (**C**) NACE.

**Figure 3 ijms-23-04549-f003:**
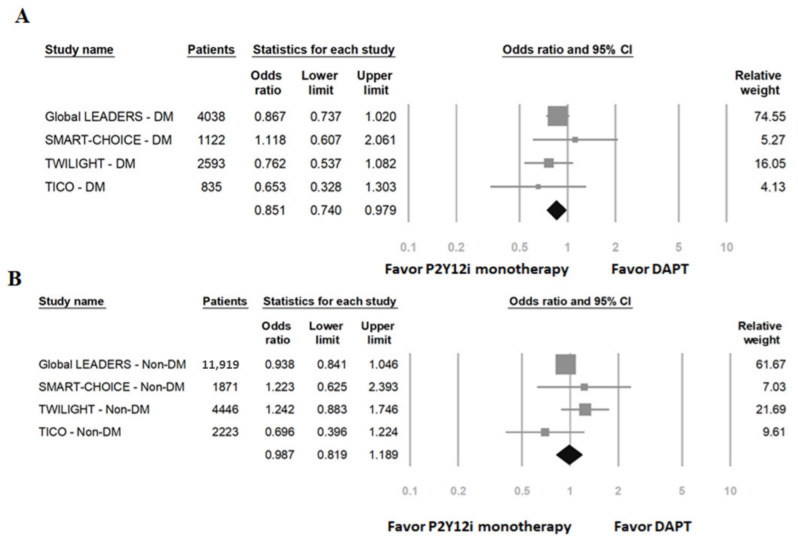
The primary efficacy outcomes (a composite of major adverse cardiovascular and cerebrovascular events) of P2Y12 inhibitor monotherapy compared to 12-month DAPT. (**A**) Patients with DM; (**B**) patients without DM.

**Figure 4 ijms-23-04549-f004:**
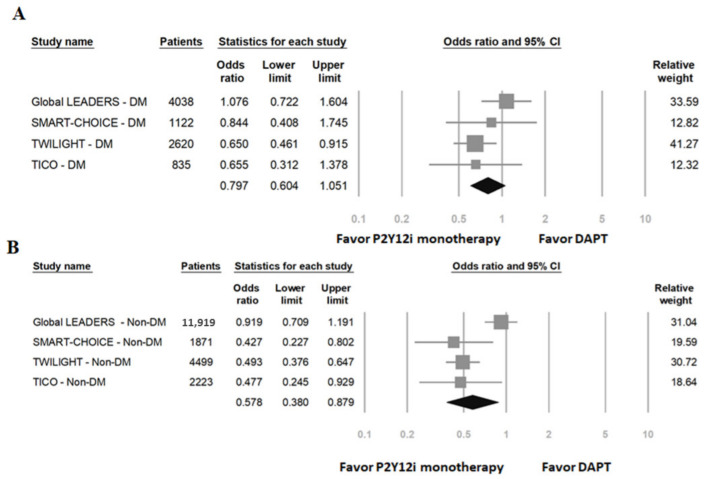
The primary safety outcomes (major or minor bleeding events) of P2Y12 inhibitor monotherapy compared to 12-month DAPT. (**A**) Patients with DM; (**B**) patients without DM.

**Table 1 ijms-23-04549-t001:** Clinical characteristics and outcomes of included randomized trials.

Clinical Trials	Global Leaders [2]	Global Leaders [2]	Smart-Choice [3]	Smart- Choice [3]	Twilight [4]	Twilight [4]	Tico [5]	Tico [5]
Year	2018	2018	2019	2019	2019	2019	2020	2020
Study population	PCI	PCI	PCI	PCI	High-risk, PCI	High-risk, PCI	ACS, PCI	ACS, PCI
Arm	DAPT 1 m, then mono	DAPT	DAPT 3 m, then mono	DAPT	DAPT 3 m, then mono	DAPT	DAPT 3 m, then mono	DAPT
P2Y12 inhibitor	Ticagrelor	Ticagrelor or clopidogrel	Clopidogrel (77%)	Clopidogrel (77%)	Ticagrelor	Ticagrelor	Ticagrelor	Ticagrelor
Patients number	7980	7988	1495	1498	3555	3564	1527	1529
Age (mean)	64.5	64.6	64.6	64.4	65.2	65.1	61	61
ACS (%)	3750 (47.0)	3737 (46.8)	870 (58.2)	873 (58.2)	2273 (63.9)	2341 (65.7)	1527 (100)	1529 (100)
STEMI (%)	1062 (13.3)	1030 (12.9)	164 (11.0)	150 (10.0)	Excluded	Excluded	546 (35.7)	557 (36.4)
NSTEMI (%)	1684 (21.1)	1689 (21.1)	239 (16.0)	230 (15.4)	1024 (28.8)	1096 (30.8)	539 (35.3)	488 (31.9)
DM (%)	2049 (25.7)	1989 (24.9)	570 (38.2)	552 (36.8)	1319 (37.1)	1301 (36.5)	418 (27.4)	417 (27.2)
Follow-up time	24 m	24 m	12 m	12 m	12 m	12 m	12 m	12 m
Primary endpoint	Death, new Q-wave MI	Death, new Q-wave MI	death, MI, stroke	death, MI, stroke	Bleeding	Bleeding	NACE	NACE
MACCE (%)	407 (5.10)	421 (5.27)	42 (2.9)	36 (2.5)	135 (3.9)	137 (3.9)	35 (2.3)	51 (3.4)
All-cause death at 12 m (%)	108 (1.35)	131 (1.64)	21 (1.4)	18 (1.2)	34 (1.0)	45 (1.3)	16 (1.1)	23 (1.5)
CV death at 12 m (%)	N/A	N/A	11 (0.8)	13 (0.9)	26 (0.8)	37 (1.1)	7 (0.5)	12 (0.8)
MI at 12 m (%)	179 (2.24)	158 (1.98)	11 (0.8)	17 (1.2)	95 (2.7)	95 (2.7)	6 (0.4)	11 (0.7)
Stroke (%)	52 (0.65)	49 (0.61)	11 (0.8)	5 (0.3)	16 (0.5)	8 (0.2)	8 (0.5)	11 (0.7)
Stent thrombosis ‡	53 (0.66)	41 (0.51)	3 (0.2)	2 (0.1)	14 (0.4)	19 (0.6)	6 (0.4)	4 (0.3)
Major or minor bleeding #	529 (6.63)	532 (6.66)	28 (2.0)	49 (3.4)	141 (4.0)	250 (7.1)	53 (3.6)	83 (5.5)
Major bleeding #	117 (1.47)	136 (1.70)	12 (0.8)	14 (1.0)	34 (1.0)	69 (2.0)	25 (1.7)	45 (3.0)
NACE	616 (7.72)	653 (8.17)	65 (4.5)	81 (5.6)	163 (4.6)	196 (5.5)	59 (3.9)	89 (5.9)

‡ Stent thrombosis was defined as definite or probable thrombosis, according to the Academic Research Consortium. # The bleeding outcome was defined according to TIMI criteria in TICO study, and BARC criteria in GLOBAL LEADERS, SMART-CHOICE, and TWILIGHT study. Major bleeding was defined as BARC type 3-5 bleeding, and major or minor bleeding was BARC type 2-5 bleeding. Values are *n*(%) unless otherwise indicated. ACS: acute coronary syndrome; BARC: Bleeding Academic Research Consortium; CV: cardiovascular; DAPT: dual antiplatelet therapy; DM: diabetes mellitus; m: month; MACCE: major adverse cardiovascular and cerebrovascular events; MI: myocardial infarction; NACE: net adverse clinical events; N/A: not applicable; NSTEMI: non-ST-elevation myocardial infarction; PCI: percutaneous coronary intervention; STEMI: ST-elevation myocardial infarction; TIMI, thrombolysis in myocardial infarction.

**Table 2 ijms-23-04549-t002:** The efficacy and safety outcomes of P2Y12 inhibitor monotherapy in patients with and without diabetes mellitus of the included randomized studies.

	DM Patients				Non-DM Patients			
	P2Y12i Monotherapy	DAPT	Hazard Ratio (95% CI)	*p*-Value	P2Y12i Monotherapy	DAPT	Hazard Ratio (95% CI)	*p*-Value
GLOBAL LEADERS	*n* = 4038				*n* = 11,919			
MACE	338 (16.7)	369 (18.7)	0.87 (0.74–1.02)	0.09	711 (12.2)	761 (12.8)	0.94 (0.84–1.05)	0.25
Bleeding	52 (2.6)	47 (2.4)	1.08 (0.72–1.60)	0.72	111 (1.9)	122 (2.1)	0.92 (0.71–1.19)	0.52
SMART-CHOICE	*n* = 1122				*n* = 1868			
MACE	23 (4.1)	20 (3.8)	1.12 (0.61–2.06)	0.72	19 (2.1)	16 (1.7)	1.22 (0.63–2.29)	0.56
Bleeding	14 (2.6)	16 (3.0)	0.84 (0.41–1.75)	0.65	14 (1.6)	33 (3.6)	0.43 (0.23–0.80)	0.01
TWILIGHT	*n* = 2620				*n* = 4499			
MACE	59 (4.6)	75 (5.9)	0.76 (0.54–1.08)	0.13	76 (3.5)	62 (2.8)	1.24 (0.88–1.75)	0.21
Bleeding	58 (4.5)	86 (6.7)	0.65 (0.46–0.91)	0.01	83 (3.8)	164 (7.3)	0.50 (0.39–0.66)	<0.01
TICO	*n* = 835				*n* = 2221			
MACE	14 (3.4)	21 (5.1)	0.65 (0.33–1.30)	0.23	21 (1.9)	30 (2.7)	0.70 (0.40–1.22)	0.21
Bleeding	12 (2.9)	18 (4.5)	0.66 (0.31–1.38)	0.26	13 (1.2)	27 (2.4)	0.48 (0.24–0.93)	0.03

Values are *n* (%) unless otherwise indicated. MACE: Major adverse cardiovascular events; P2Y12i: P2Y12 inhibitor.

## Data Availability

Data available in a publicly accessible repository. The data presented in this study are openly available in PubMed, Embase, Cochrane library database, ClinicalTrials.gov (accessed on 23 February 2022), and websites including www.escardio.org, www.acc.org/cardiosourceplus, www.tctmd.com (accessed on 23 February 2022).

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
