# Peer review of "Comparative Efficacy and Safety of P2Y12 Inhibitor Monotherapy and Dual Antiplatelet Therapy in Patients with and without Diabetes Mellitus Undergoing Percutaneous Coronary Intervention"

_ijms, 2022, doi:10.3390/ijms23094549_

Round 1

Reviewer 1 Report

The manuscript is interesting and it focuses on an important topic. Several studies highlight the efficacy of shorter DAPT therapy in order to reduce the risk of bleeding, without and incresing risk of a new ischemic event. This work fits in this continuum, focusing, in particular, on diabetic patients. However i have some suggestion to improve the manuscript:

1) Endothelial dysfunction is a condition often associated with coronary artery disease and microvascular dysfunction. Endothelial dysfunction is closely associated with cardiovascular risk factors as diabetes mellitus, despite it is also described as an independent determinant of ischemic heart disease and acute coronary syndrome (see  J Cardiovasc Dev Dis. 2021 Sep 18;8(9):116. doi: 10.3390/jcdd8090116). Please add a brief discussion regarding the role of endothelial dysfunction and possible involvement in the response to DAPT in patients  with ischemic heart disease (see  Int J Mol Sci. 2022 Mar 18;23(6):3301. doi: 10.3390/ijms23063301. PMID: 35328719).

2) Please discuss the timing of antiplatelets suspension and possible differences in term of events occurence.

3) please define all the abbreviations first time they appear in the main text, abstract, tables

4) fine english editing is required

Author Response

Response to Reviewer 1 Comments

1) Endothelial dysfunction is a condition often associated with coronary artery disease and microvascular dysfunction. Endothelial dysfunction is closely associated with cardiovascular risk factors as diabetes mellitus, despite it is also described as an independent determinant of ischemic heart disease and acute coronary syndrome (see  J Cardiovasc Dev Dis. 2021 Sep 18;8(9):116. doi: 10.3390/jcdd8090116). Please add a brief discussion regarding the role of endothelial dysfunction and possible involvement in the response to DAPT in patients  with ischemic heart disease (see  Int J Mol Sci. 2022 Mar 18;23(6):3301. doi: 10.3390/ijms23063301. PMID: 35328719).

Response:

The discussion of endothelial dysfunction in DM patients is added to our manuscript on page.9, and these two articles were added to our reference.

"DM also leads to endothelial dysfunction, one of the main pathophysiologic mechanisms associated with cardiovascular disease and is described as an independent determinant of ischemic heart disease and acute coronary syndrome.[24] Several biochemical pathways have been described to demonstrate the association between endothelial dysfunction and platelet activation, such as nitric oxide (NO) and prostacyclin (PGI2).[25] NO, a well-known atheroprotective and vasodilating substance, may also attenuate platelet activation. Blood vessels of patients with DM have diminished NO production, enhanced NO degradation, and decreased sensitivity to NO. PGI2 is another important regulator produced by endothelial cells that inhibits platelet activation through binding to the prostacyclin receptor on platelets. DM is associated with lower levels of prostacyclin synthase in subcutaneous arteries and possibly leads to impaired formation of PGI2.[26] Together, the endothelial dysfunction and platelet hyperactivity make DM patients much more susceptible to cardiovascular disease than non-DM patients.[27]"

2) Please discuss the timing of antiplatelets suspension and possible differences in term of events occurrence.

Response:

The discussion of suspension of antiplatelets and possible differences in term of events occurrence outcomes was added to our manuscript on page 8.

"Conventionally, it is recommended that DAPT should be continued for at least 6 months (in stable CAD) or 12 months (in acute coronary syndrome) unless contraindications occur.[12] Early suspension of antiplatelet therapy would increase the risk of recurrent ischemic events and stent thrombosis. This concept was changed because of the advent of safer newer-generation DES and the awareness of increased bleeding risk caused by prolonged DAPT. Based on these reasons, a new treatment strategy of using a potent P2Y12 inhibitor instead of DAPT was proposed."

3) please define all the abbreviations first time they appear in the main text, abstract, tables

Response:

We proofread the manuscript again and added definitions to all abbreviations the first time they appeared.

4) fine English editing is required

Response:

We will use the English editing services in MDPI to polish our manuscript. (https://www.mdpi.com/authors/english) 

We would like to express our gratitude to your kind suggestions to our manuscript. We revised the paper as far as we can and hope it can achieve the standard of publication in “Int J Med Sci ”. Thanks again for your kind help.

Reviewer 2 Report

Dear Authors

            The article “Comparative efficacy and safety of P2Y12 inhibitor monotherapy and dual antiplatelet therapy in patients with and without diabetes mellitus undergoing percutaneous coronary intervention” is a very important voice in the discussion on antiplatelet therapy following PCI with stent implantation. It seems particularly important to draw the authors' attention to the differences in the response of patients with and without diabetes.
            The P2Y12 inhibitor monotherapy in patients without diabetes  significantly reduced the risk of ischemic events  and also the risk of bleeding complication. In contrast, in patients with diabetes monotherapy reduced  the risk of ischemic events but not the risk of bleeding.
            However, I think that the Authors should pay more attention in the discussion to the fact interindividual variability in the response to oral antiplatelet drugs. In the aspect of monotherapy, this applies primarily to Clopidogrel, which is a prodrug and requires metabolism to the active form. Metabolic disorders as well as competition for CYP3A4 with other drugs (e.g. statins) may lead to a lower active drug concentration and to insufficient antiplatelet effect. So, in my opinion, the thesis that P2Y12 inhibitor monotherapy shows superiority over the dual antiplatelet therapy should be limited to potent P2Y12 inhibitors.

Author Response

Response to Reviewer 2 Comments

Dear Authors

    The article “Comparative efficacy and safety of P2Y12 inhibitor monotherapy and dual antiplatelet therapy in patients with and without diabetes mellitus undergoing percutaneous coronary intervention” is a very important voice in the discussion on antiplatelet therapy following PCI with stent implantation. It seems particularly important to draw the authors' attention to the differences in the response of patients with and without diabetes.
   The P2Y12 inhibitor monotherapy in patients without diabetes significantly reduced the risk of ischemic events and also the risk of bleeding complication. In contrast, in patients with diabetes monotherapy reduced the risk of ischemic events but not the risk of bleeding.
    However, I think that the Authors should pay more attention in the discussion to the fact interindividual variability in the response to oral antiplatelet drugs. In the aspect of monotherapy, this applies primarily to Clopidogrel, which is a prodrug and requires metabolism to the active form. Metabolic disorders as well as competition for CYP3A4 with other drugs (e.g. statins) may lead to a lower active drug concentration and to insufficient antiplatelet effect. So, in my opinion, the thesis that P2Y12 inhibitor monotherapy shows superiority over the dual antiplatelet therapy should be limited to potent P2Y12 inhibitors.

Response:

The discussion of impaired response to clopidogrel in DM patients is added to our manuscript on page.9

"Besides, clopidogrel is a prodrug that requires metabolism to transform into active form. DM patients have a greater prevalence of nonresponders to clopidogrel than non-DM patients.[33] Impaired drug metabolism, metabolic disorders, and competition for CYP3A4 with other drugs (e.g., statins) are possible mechanisms leading to reduce concentration of clopidogrel's active metabolite and insufficient antiplatelet effect.[34,35]"

We would like to express our gratitude to your kind suggestions to our manuscript. We revised the paper as far as we can and hope it can achieve the standard of publication in “Int J Med Sci ”. Thanks again for your kind help.
